# Automatically Advancing LLM Expertise in Technology Judgment

## Abstract

Large language models (LLMs) are rapidly becoming core tools for science, engineering, and innovation. Their promise lies not just in remembering facts, but in putting knowledge to work. Despite their impressive ability to answer increasingly difficult questions, it remains unclear whether LLMs truly use their knowledge when confronted with new and challenging tasks. We address this question with a patent classification task that requires deep conceptual understanding: distinguishing objectively different but semantically similar patents. To evaluate this approach, we introduce a challenging new benchmark of 1.3 million post-2015 computer science patent pairs, characterized by dense technical jargon and strategically complex writing. We find that LLMs often fail our benchmark and struggle to distinguish among semantically similar patents. To probe this failure, we introduce a novel framework that decomposes model errors into two sources: missing and unused knowledge. Our approach asks models to generate clarifying questions to improve their understanding, and then compares three settings: raw performance, self-answered questions, and externally supplied answers. This decomposition reveals that LLMs often possess the relevant knowledge internally but fail to deploy it, while a smaller share of errors arises from genuine knowledge gaps. We then ask whether the ability of models to construct a task-specific database of questions and answers differs across models. We find that smaller models generate simpler, broadly transferable questions, while larger models propose more complex but less generalizable ones. This suggests new strategies for combining strengths across models. *Taken together*, our findings highlight a critical limitation of current LLMs and their evaluation: models often know more than they can use. By shifting evaluation from recall of static facts to application of dynamic knowledge, our approach provides a more informative lens on model capabilities and opens a path toward building systems that better support technological discovery and innovation.

## 1 Introduction

Large language models (LLMs) are increasingly understood not merely as passive text generators, but systems capable of reasoning about and engaging with complex realities. Recent studies have highlighted their potential for self-improvement—forming hypotheses, seeking clarification, and revising beliefs through iterative interactions (Huang et al., 2023; Qu et al., 2024; Song et al., 2024; Wang et al., 2023c;b). Yet, despite their impressive capabilities, LLMs often struggle with tasks that hinge on subtle conceptual distinctions (Asthana et al., 2024; Havlík, 2024; Pavlick, 2023). Much of their internal knowledge remains latent, poorly organized, and difficult to access (Pan et al., 2025; Hoang et al., 2023). This reflects a fundamental tension: while LLMs appear to "know everything," their knowledge is stored in a heavily compressed form that impairs activation, recall, and information quality assessment (Hoang et al.; 2023; Jaiswal et al., 2024).

In this paper, we ask whether LLMs can enhance their own understanding by acting more like learners: generating and answering their own questions – often tailored to their specific uncertainties and usage, reflecting personalized internal gaps rather than universally missing facts – to identify and fill in the background knowledge required for task completion. This simple strategy of self-questioning, especially when paired with external retrieval, can automatically improve judgment, expose internal knowledge gaps, and activate reasoning processes that not only enhance the model's own performance

but also transfer effectively to larger models, where such reasoning often remains dormant under standard prompting.

A further challenge we highlight is the distinction between *lay-in knowledge* and *working knowledge*. In humans, once a fact is known, it is typically accessible for reasoning and problem solving: retrieval and use are closely aligned. In LLMs, however, the presence of knowledge in parameters does not guarantee its availability in practice. Prior work has probed this issue by designing tasks that isolate narrow domains (e.g., math word problems or factual recall) to demonstrate what knowledge LLMs are or are not using (Asthana et al., 2024; Havlík, 2024; Pavlick, 2023). By contrast, our framework is situated in a broad technological domain, where the challenge is not the absence of information but the difficulty of activating and applying it in context. This generalizable setup, built on a large and diverse patent corpus, goes beyond special-case demonstrations. It allows us to investigate systematically how LLMs bridge the gap between latent stored knowledge and the working knowledge required for fine-grained conceptual judgment.

Our *first contribution* is the proposal of **a series of behavioral experiments to systematically evaluate self-questioning** as a core mechanism for active learning and introspective growth in LLMs. We focus on the model's understanding of technology concepts, operationalized through a challenging pairwise differentiation task. Instead of assessing understanding through conventional summarization or question-answer (QA) tasks, we present LLMs with pairs of semantically similar but conceptually distinct patent abstracts and ask whether they describe the same invention. A model that truly understands the material must go beyond surface-level similarity and detect subtle functional or mechanistic differences. To rigorously test this ability, we construct **a new dataset of over 1.3 million close-but-distinct computer science patent pairs**, granted post-2015 by the United States Patent and Trademark Office (USPTO), which is our *second contribution*. Patents present a uniquely demanding testbed: they are written in dense, legal-technical language, often strategically obfuscated, and their distinctions are adversarially verified by expert patent examiners. Our task requires fine-grained conceptual discrimination that goes well beyond pattern matching, skills, imitation, or memorization that could possibly bypass the existing benchmarks (McIntosh et al., 2024; Wu et al., 2024; Davis, 2023) [1]. We embed a structured self-questioning phase between initial exposure and final prediction. Given a pair of closely related patents, the model generates questions targeting both surface-level details and deeper conceptual understanding. It retrieves relevant scientific content to answer its own questions, combining external evidence with internal knowledge. These QA pairs, without verified correctness, are then fed back into the model to inform its final judgment on whether the patents describe the same invention. This setup allows us to assess whether and how self-generated inquiry and targeted retrieval enhance the model's conceptual understanding (i.e., its accuracy).

Using our framework, we show that prompting LLMs with self-questioning significantly improves their performance on the patent differentiation task, enhancing judgment accuracy, confidence, and response consistency. We then explore three hypotheses to understand what drives this improvement. One possibility is that the questions themselves help the model parse the patent text more effectively by serving as informative cues. Another is that LLMs possess relevant internal knowledge but often fail to access it. A third hypothesis is that even when the model does retrieve relevant knowledge, its internal representations may compress and lose useful information, such that access to external science demonstrably improves performance. To test these possibilities, we design experiments to explore **how LLMs' internal and external knowledge are organized and transferred across model scales**, especially when they are sparsely distributed like science, which is our *third contribution*. We compare the model's accuracy when given only the questions, when using its questions and self-generated answers, and when using questions and answers derived from science. We find that prompting the model to ask itself questions helps it parse the patent text and improves performance. Feeding LLMs with their self-produced answers also helps, suggesting that LLMs often fail to retrieve relevant internal knowledge. We also find that self-generated answers are less effective than externally sourced ones, however, indicating that hidden compressed capacity remains insufficient for complex judgment tasks, where raw, external scientific knowledge in the form of QA proves beneficial to add richness. Interestingly, we find that small models often generate more useful, open-ended, and better-aligned questions for mid-sized models than larger models do, which highlights the importance of matching question complexity to model capacity, as well as a scalable prompting strategy where

---

[1]We have more detailed discussions in Appendices A and B.

smaller models help structure the problem in a straightforward way for stronger models to reason through.

## 2   A UNIQUE DATASET OF HARD-TO-DISTINGUISH PATENTS

Understanding is a complex cognitive ability that goes beyond recalling facts or executing procedures. It involves grasping the deep relational structures that connect concepts, including causal links, analogies, and hierarchies. Drawing from cognitive science and philosophy, a central aspect of understanding is differentiation: recognizing what makes two concepts distinct. This ability is crucial for constructing models of conceptual relationships and is especially vital for causal reasoning, where detecting contrasts and similarities supports counterfactual thinking and explanatory modeling.[2]

Because direct questions like "what is X?" often elicit superficial or memorized responses, we need a more robust test of understanding. Instead, we propose a task that evaluates the model's capacity to discriminate between concepts that are linguistically similar but semantically distinct. Our test centers on pairwise differentiation: the model is presented with two short texts and asked whether they describe the same underlying idea or two distinct ones. These texts are carefully selected to be semantically similar in language and structure — making the task deceptively difficult, and surface-level heuristics ineffective. The model must detect subtle differences in purpose, mechanism, and/or function, and reason about their implications.

We apply this task in the technology domain and focus specifically on patent abstracts. *We construct a novel dataset consisting of 1,319,184 pairs of information and computer technology patent abstracts granted by the USPTO after 2015*. In the Appendix C, we also explore another type of "merging" task: whether a model can recognize a rewritten description referring to the same invention. We find that the differentiation task requires genuine conceptual linking and functional comparison, making it a strong testbed for evaluating LLM understanding in the technology domain. Several features make technology patents particularly well-suited for testing fine-grained semantic distinctions:

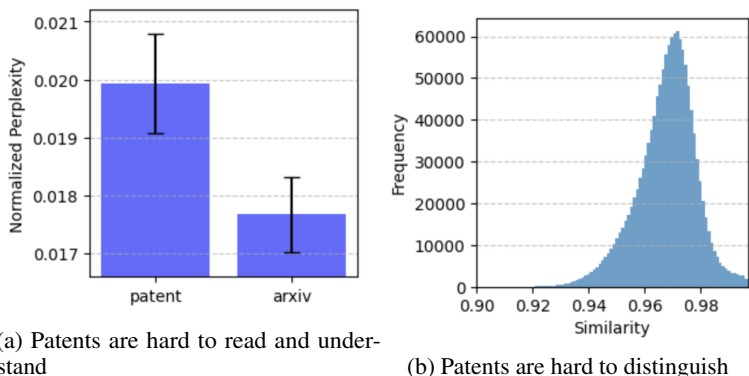

(a) Patents are hard to read and understand

(b) Patents are hard to distinguish

Figure 1: Panel (a): Model perplexity of LLaMA-3.1-8B-Instruct on paper and patents abstracts. Panel (b): Histogram of all 1,319,184 patents, showing the distribution of each patent's maximum cosine similarity to another (distinct) patent.

**First**, patents often describe components of larger systems and therefore **exist in dense clusters of near-duplicate-documents** that are nearly identical in structure and language but differ in one or two meaningful ways, such as a specific constraint, use case, or technical improvement (Trippe, 2015). These subtle distinctions have been rigorously evaluated by human judges (we will develop this in the third point) and represent precisely the type of contrasts that test whether an LLM can move beyond surface-level similarity and recognize functional difference. We illustrate this pattern in Figure 1b, which shows the distribution of the most similar—but non-identical—patent pairs based on textual embeddings from (Ghosh et al., 2024), pretrained on patent-citation relations. The

---

[2]Section A in the appendix offers a more detailed discussion of the link between understanding and pairwise differentiation.

distribution is heavily concentrated at high similarity scores, suggesting that most patents have at least one semantically similar counterpart.

**Second**, patent documents are often **strategically obfuscated**, written to disclose the minimum required for legal protection while revealing as little as possible about the underlying invention. As a result, their language tends to be intentionally dense, formal, and legally contorted. Readability studies show that patent specifications—particularly the claims sections—require significantly higher grade-level comprehension than comparably technical scientific prose, due to their use of dense jargon and legalistic drafting norms (Kong et al., 2023; Ouellette, 2012). This linguistic complexity is further illustrated in Figure 1a, which compares the model perplexity of LLaMA-3.1-8B-Instruct on 4,000 randomly selected computer science paper abstracts from ArXiv and 4,000 matched patent abstracts from the same years. Perplexity, while not a direct measure of understanding, reflects the model's inability to predict the next word in a sequence. Consistently higher perplexity on patent abstracts suggests that they contain highly technical and domain-specific language that poses a greater challenge for LLMs to parse and comprehend. An example of paired patent abstracts and a related paper abstract can be seen in Appendix D.

**Third**, and most importantly, even though patents may seem similar, we know they are not. Each granted patent in our dataset has undergone an **extensive expert evaluation process conducted by patent examiners** at the USPTO. This review assesses whether the application satisfies the legal requirements of novelty, non-obviousness, and utility. During this process, examiners conduct detailed prior art searches, compare the proposed invention to existing patents and publications, and determine whether the claimed invention constitutes a meaningful and non-trivial contribution over what already exists. This institutional validation is crucial for our task: it provides a strong, domain-informed guarantee that two different granted patents indeed describe conceptually distinct inventions.

# 3 EVALUATING LLMS' UNDERSTANDING OF THE TECHNOLOGY DOMAIN

We now describe how we operationalize the pairwise differentiation task. We begin by sampling pairs of patents that are difficult to distinguish, identified by embedding all patent abstracts using PATCER (Ghosh et al., 2024), a contrastive learning model trained on patent citation relationships. For each patent, we select its most similar—but non-identical—counterpart based on cosine similarity in the embedding space. These pairs are then presented to the LLM, which is asked whether the two abstracts describe the same invention.[3] For each prompt, the model returns two outputs: a binary label indicating "yes" (1) or "no" (0), and a confidence score on a scale from 0 to 10.

To construct our main measure of understanding, we aggregate predictions using a confidence-weighted voting scheme applied over three independent generations of the same prompt. If the total confidence associated with label "1" exceeds that of label "0," the final label is set to 1; otherwise, it is 0. This aggregation method has been shown to represent the model's actual certainty, accelerate convergence toward the model's stable judgment, and improve self-consistency relative to simple majority voting (Geng et al., 2024). In addition to providing more reliable final decisions, confidence scores offer a fine-grained signal that helps distinguish between borderline and clear-cut cases. This is particularly important in our setting, where many patent pairs are deceptively similar—confidence levels can help identify when the model is genuinely uncertain (Geng et al., 2024). Alongside final labels, we also track changes in confidence and measure self-consistency, defined as the proportion of times all three responses yield the same answer. Lower consistency indicates greater uncertainty in model judgment.

Figure 2 illustrates how models of different sizes perform on the differentiation task within the technology domain. The first panel shows that smaller models, such as LLaMA-8B, struggle to distinguish between semantically similar but distinct patents, classifying over 70% of patent pairs as the same. In contrast, the larger LLaMA-405B model labels only around 10% of these pairs as identical, indicating a stronger ability to detect subtle distinctions. The second and third panels show that the smaller model is also less confident and consistent in its responses (across blue bars). The 8b model's confidence and self-consistency of wrong answers (blue) is way higher than those of the right answer (orange), again suggesting higher uncertainty and less stable internal representations compared

---

[3]See Appendix E for full prompts.

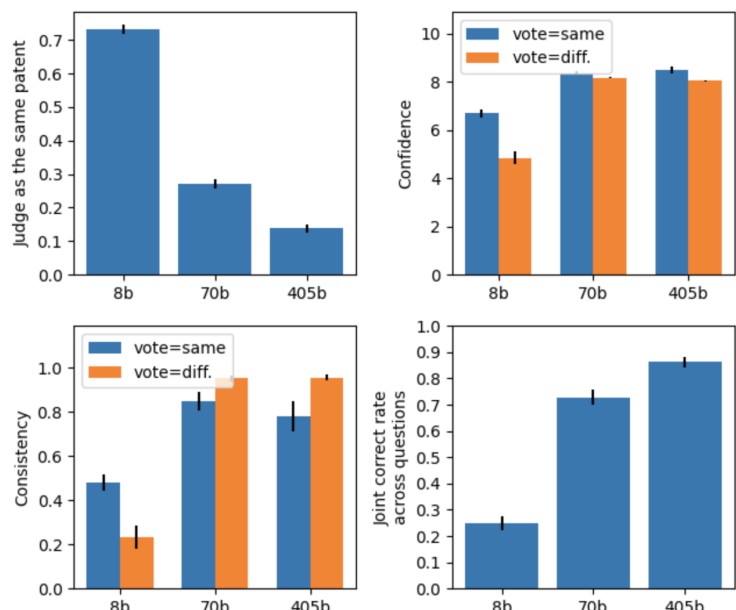

Figure 2: Ask LLMs "Do you think they are the same patent/different patents?", given that they are actually different. All models are Llama 3.1 instruction-tuning versions.

to the larger model. The fourth panel evaluates this phenomenon directly by measuring whether the model gives consistent answers across two question forms. For each patent pair, we ask both "Do you think these patents describe the same invention?" and "Do you think they describe different inventions?" A model with a stable internal representation of the concepts should give logically consistent answers across these two formulations, measured by joint-correct rate, the proportion of pairs where the model gives both correct answers. Deviations from this indicate sensitivity to surface-level wording. The results again show that smaller models are less consistent and more affected by such superficial cues in the phrasing, reinforcing the idea that they lack a robust conceptual understanding of the task.

In the appendix F.1, we show that across all LLaMa models, responses shift systematically depending on how the question is framed. All models are more likely to judge the patents as different when asked explicitly whether they are different, even though the content of the question is logically equivalent. We also report additional results for other model families with varying sizes, including Qwen, Mixtral, and GPT-4o-mini. Again, we find that larger models perform better on the differentiation task. Interestingly, we observe consistent response biases in some models: Mixtral models tend to answer "yes" regardless of the question's phrasing, while Qwen models tend to answer "no." Based on these findings, we select the LLaMA-3.1-8B Instruct model as the primary focus for our subsequent experiments and use the question "Do you think they are the same patent?" as the default formulation for the pairwise differentiation task.

## 4 SELF-QUESTIONING LLMS FOR IMPROVED TECHNOLOGY UNDERSTANDING

We begin by examining whether prompting models to ask themselves structured questions can enhance performance and, if so, how. We design two types of self-generated questions. The first, surface-level questions, target conceptual recall (e.g., What is X? What does Y do?), while the second encourages deeper understanding through questions about the integration of concepts, improvement, and the intent underlying invention. We construct the pipeline illustrated in Figure 3. When the model is exposed to a pair of closely related patents, $P_1$ and $P_2$, it generates six questions $Q_{ij}$ for each, where $i \in 1, 2$ indexes the patent and $j \in 1, \dots, 6$ indexes the question. Three of these are surface-level questions, and three target deeper comprehension. These questions guide the

retrieval of relevant scientific knowledge, mimicking how researchers ask clarifying questions to deepen understanding. For external knowledge, we use parsed, pre-processed full texts from arXiv computer science papers. Using Google Scholar, we identify the top 10 papers relevant to the patents, segment each into overlapping 2500-character chunks (with 200-character overlaps), and treat these as candidate knowledge units. We embed both the questions and text chunks using the SPECTER2 model (Singh et al., 2023) specifically for scientific documents pretrained on citation relations, compute cosine similarity, and select the top three relevant chunks per question. These are then passed to the LLM as external knowledge $Knowl_{ij}$. The model answers its own questions using both its internal knowledge and the retrieved content, producing $A_{ij} = f_{LLM}(Q_{ij} \mid Knowl_{ij})$. Then the LLM makes a decision by $Decision = Judge(Patent_1, Patent_2 \mid \{(Q_{1j}, A_{1j})\}_{j=1}^{6}, \{(Q_{2j}, A_{2j})\}_{j=1}^{6})$ with the previously mentioned confidence-weighted voting schema.

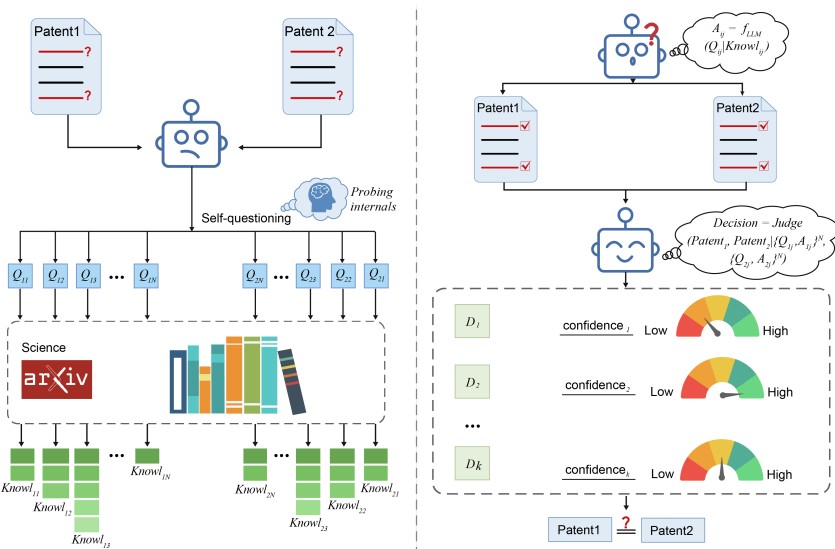

Figure 3: Self-questioning LLMs: The model probes its internals to generate questions, retrieving scientific knowledge or drawing from its own knowledge base to answer them (left). It then uses these structured question-answer pairs to revisit the task under the confidence-weighted voting schema (right).

## 5 RESULTS

In this section we decompose the error rate into its components - how much of the error is due to missing information and how much is due to the inability to retrieve self-knowledge. Section 5.1 shows our main result. In the following section, we analyze how model size affects question quality and judgment accuracy (5.2).

### 5.1 LLMs DO NOT KNOW WHAT THEY KNOW

Figure 4a presents our initial results from a random sample of 1000 patent pairs, examining how questions and answers affect classification performance. The baseline represents the initial accuracy of Llama 8B without any self-questioning or scientific retrieval. At this baseline level, Llama 8B incorrectly classifies approximately 70% of patent pairs as identical.

The blue line demonstrates the impact of adding questions alone. This modest improvement suggests that simply posing understanding questions helps the model perform better by providing structure that aids text parsing, though the effect of merely enumerating questions remains limited.

The orange line reveals that adding self-generated answers significantly improves performance. This substantial gain implies that the model already contains useful knowledge for the differentiation task, but this knowledge is not being effectively utilized. These results suggest that LLMs do not know

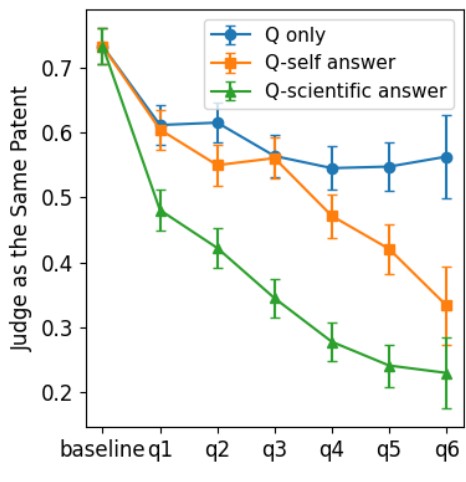 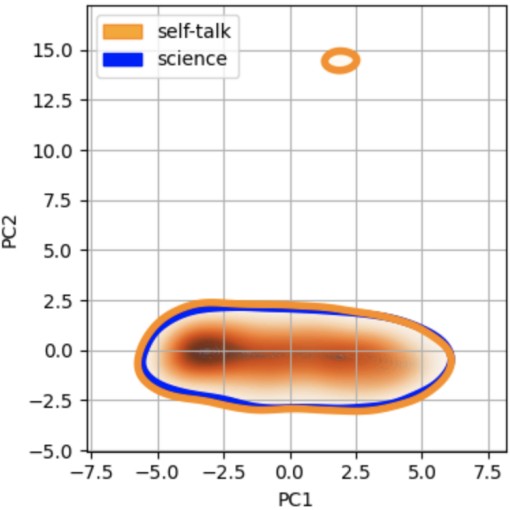

(a) LLM does not know what it has known: Self-talk helps improve judgment, but not as much as scientific QA. $p$-values: $p_{q_1}$ to $p_{q_5} = 0.000$, $p_{q_6} = 0.013$ (self vs. scientific answers).

(b) Scientific retrieval does not introduce more "content-level" new information for LLM to answer these questions.

Figure 4: Mechanisms of "LLMs do not know what they know".

what they know—bringing forth relevant information that enhances understanding can dramatically improve performance.

Finally, the green line shows the improvement achieved by providing retrieved answers. This additional performance gain indicates that while the model's internal knowledge is valuable, it does not match the quality of complete information available in the training data. Notably, with six question-answer pairs, performance reaches the same level as the 405B parameter model, effectively eliminating the error rate.

Importantly, most of the error reduction is achieved through self-generated answers, suggesting that the model contains the most useful information within its internal knowledge representation. However, the model lacks the ability to effectively access and apply this knowledge when tackling tasks that require such domain expertise.

Our results demonstrate that models fail to retrieve the information necessary for the task, and the information they do retrieve is of lower quality than that obtained from additional sources. We next explore the factors driving this performance difference, considering two possible explanations. First, LLMs function as compressed representations of knowledge but do not preserve complete information. Scientific and technological information tends to be sparsely and unevenly distributed in training corpora, and the compression process significantly degrades the fidelity of such underrepresented knowledge. Second, LLMs may acquire new knowledge at inference time—meaning the external knowledge we provide is entirely unknown to the LLM and must be learned and incorporated during inference.

We argue that the first explanation is likely the primary cause of the improvement we observe. The models have likely already encountered the data we use: Llama 8B was released in July 2024[4], when approximately 81% of the retrieved papers that generated scientific answers had already been published. However, while the models may have seen these papers during training, this exposure might not have significantly influenced the learned weights.

To further investigate what type of information is contained in inference-time retrieval versus the model's internal knowledge, we analyze the answers by embedding all responses from both scientific sources and the LLM's self-generated answers using the model from (Ghosh et al., 2024). To visualize and compare the distribution of embedding spaces, we reduce dimensionality to 2D and estimate

---

[4]https://huggingface.co/meta-llama/Llama-3.1-8B-Instruct

boundaries using 5% Gaussian kernel density estimation—drawing contours that enclose the densest 95% of data points in each embedding set.

If LLMs substantially incorporate new scientific content during inference, we would expect their embedding regions to diverge meaningfully from those derived from self-generated responses. However, as shown in Figure 4 panel b, the embedding regions largely overlap. Notably, self-generated responses display a distinct outlier region, which we suspect reflects hallucinated content introduced by the LLM.

Despite occupying similar regions in the 2D embedding space that captures the greatest variance in the high-dimensional (768D) space, these two answer types differ in several meaningful ways. Scientific answers average $62.1 \pm 1.8$ words and are presumably richer in detail and informativeness, drawing more heavily from scientific evidence. In contrast, self-generated answers are notably shorter, averaging only $38.2 \pm 0.6$ words—approximately half the length of scientific responses. We therefore, interpret the results as suggesting that although the LLMs already have the knowledge needed to answer the question, and obtain better performances - this knowledge is compressed, less detailed and not as rich as the original scientific data.

## 5.2 It's Not Only What to Ask, but who asks it

Self-questioning helps because it aligns the model's attention with task-relevant concepts. This suggests a transferring question: are the most useful questions those that *the same* model would ask itself, or can questions authored by a different model—smaller or larger—better align the answerer's knowledge with the task? We next test how question–answer sets travel across scale and where misalignment helps or hurts.

The answer depends on which model is doing the answering. We test two models—LLaMA 3B and 8B—on questions generated by four others: LLaMA 1B, 8B, 70B, and 405B. We select 3B and 8B as answerer models because the 1B model produces low-confidence responses (94.4% with a confidence score of 0), indicating little understanding of content. In contrast, the 70B and 405B models already achieve strong performance without an external scientific context, leaving little room for improvement. The 3B and 8B models provide a more informative middle ground: they show some understanding but still benefit from superior prompts.

As shown in Figure 5 panels a and b, we find that the 3B model performs best when answering questions from the 1B model, and the 8B model benefits from questions posed by both the 1B and 8B. This asymmetry reveals a critical insight: questions from larger models do not consistently help smaller models, and often introduce confusion. A more knowledgeable model cannot fully substitute for a smaller model's understanding, suggesting that knowledge extraction is not smoothly transferable across model scales (Li et al., 2024). At the same time, we observe that questions from smaller models often align well with the answering capacity of larger models. This may be because larger models can readily interpret, build on, and give a better answer to smaller models' questions. Large model questions appear to go beyond the bounds (and dimensionality) of smaller model understanding and answering capacity.

To probe this further, we generate 10,000 question-scientific answer pairs using the 3B and 8B models (the answer model), based on questions from either smaller or larger models relative to the answer models' own size (smaller or bigger question models), resulting in 40,000 total pairs. We then measure textual similarity between each QA pair and its associated patent abstract. When a larger model answers questions from a smaller model, the resulting QA content is consistently less similar to the original patent abstract, suggesting that small models' questions bring in more external scientific information to improve the quality, texture, and richness of knowledge beyond the raw patent texts (Figure 5 panels e and f). Small model questions "discover more" **external** information from the sources they search. We also re-feed the scientific QA pairs into the 3B and 8B models and record their model perplexity. This lower similarity predictably activates more **latent** reasoning in the models, as reflected in their answer's higher perplexity (Figure 5 panels c and d). This may stem from how smaller models phrase their questions. With limited capacity, their questions tend to have two important features: (1) Small model questions are more fundamental — 68% of 1B's questions contain "what is" or "what are" (usage/purpose of components, concepts, and so on), compared to only 44% for 405B. (2) Smaller models tend to ask shorter, less contrastive, and more open-ended questions, potentially yielding higher information entropy (Figure 5 panels g and h). A 1B model's

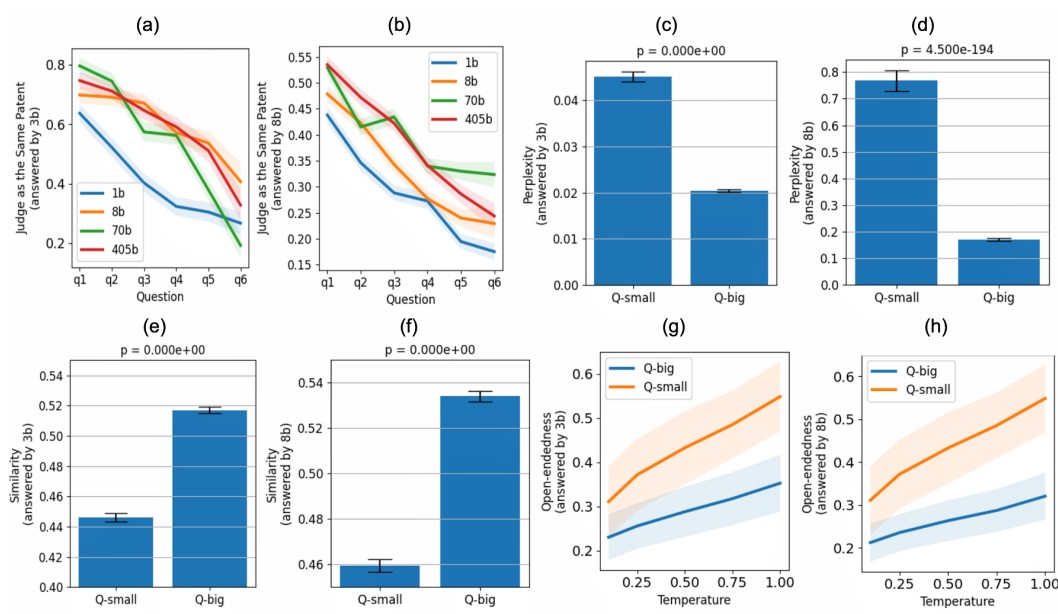

Figure 5: Panels a and b: Questions generated by smaller models are compatible with larger models, but not the converse. Panels c to h: mechanisms. Panels c and d: higher perplexity when answering small models' questions. Panels e and f: more new information when answering small models' questions. Panels g and h: smaller models' questions are more open-ended.

questions average 16.7 words, compared to 23.0 for a 405B model. We further sampled 200 questions from both smaller and larger question models. The answer model (3B or 8B) was prompted to respond to each question ten times across temperatures from 0.1 to 1, controlling for the answer model's underlying determinism. We quantify "open-endedness" as the embedding distance of each answer from the mean of all answers (standard deviation). Questions from smaller question models elicit more variable answers.

## 6  FINAL REMARKS

Self-questioning not only enhances LLMs' understanding but also reveals how they organize and access internal/external knowledge. While models hold useful internal insights, these are often latent and underused. Self-questioning helps activate this hidden compressed capacity, but falls short for complex tasks. Interestingly, the fundamental questions from smaller models often align with the needs of larger models. This suggests a scalable strategy: use smaller models to structure problems and guide deeper reasoning in stronger models.

**Impact**: our work and the associated new benchmark on patent distinction offer broad insights into how AI can be used to automate the evaluation of scientific and technical contributions.

## THE USE OF LARGE LANGUAGE MODELS (LLMS)

We employed LLMs to assist with polishing the writing. All content generated or modified by LLMs was rigorously reviewed and approved by the authors.

## ETHICS STATEMENT

This work does not involve human subjects, sensitive data, or any other issues outlined in the ICLR Code of Ethics.

## REPRODUCIBILITY STATEMENT

We submitted code samples for reproducibility.

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

# Appendix

## A    UNDERSTANDING THROUGH DIFFERENTIATION

Understanding is an ability that goes beyond merely recalling isolated facts or performing procedures; Grimm (2024) it fundamentally involves grasping the relationships that connect concepts and principles within a domain. This requires both an appreciation of and the ability to uncover underlying structures, such as causal links, correlations, hierarchies, or analogical parallels, which distinguish true comprehension from sophisticated pattern matching or associative recall. Evaluating an LLM's understanding, therefore, requires methods that probe its capacity to navigate these relational structures, moving beyond tests of surface-level knowledge.

Cognitive science, from (Gibson, 1969) through (Tversky, 1977) and Gentner's structure-mapping work (Markman and Gentner, 2000) to recent relational-reasoning research (Alexander and Dumas, 2018), demonstrated that the ability to perceive, represent, and reason about crucial semantic relationships is built upon two complementary foundational processes: differentiation, contrasting, or "splitting", and generalization, merging, or "lumping". Differentiation is the capacity to distinguish concepts, entities, or situations, recognizing the features or relations that make them distinct – essentially, identifying what makes 'A' different from 'B'. Without this ability to draw meaningful distinctions, even simple relationships cannot be properly conceptualized. Generalization, conversely, involves identifying commonalities across different instances, abstracting shared properties or underlying structures, and treating distinct items as equivalent for certain purposes, such as belonging to the same category or representing the same concept. These processes are fundamental not just for forming basic concepts, but for constructing the complex relational knowledge that constitutes deep understanding in any domain.

Moreover, the ability to differentiate and generalize remains vital for constructing and refining our understanding of relationships between concepts, especially causal ones, which serve as the backbone of understanding. To explain why something occurs, we must first detect patterns: when does outcome $B$ follow event $A$, and when does it not? This requires the capacity to distinguish $A$ from alternatives like $A'$, and to notice that $B$ follows $A$ but not $A'$. Such contrasts form the basis for counterfactual reasoning, allowing us to imagine what would happen if everything about $A$ and $A'$ were held constant except for one defining feature—if changing that feature flips the outcome, it signals a likely cause. At the same time, we must also recognize when observations are similar in respects that matter. Only by treating $A$ and $A'$ as alike for relevant purposes, and then seeing whether their outcomes differ, can we meaningfully generalize and say that "$A$ causes $B$." This dual capacity—to detect when things are different and when they are effectively the same—is what underlies both the identification and the generalization of causal claims. Contemporary computational models and econometric frameworks reflect this structure explicitly, emphasizing the need for clear conceptual boundaries and controlled variation (Pearl, 2009; Lake et al., 2017; Heckman and Pinto, 2022). Developmental research similarly shows that even young children rely on these kinds of structured comparisons to infer causes (Gopnik and Wellman, 2012).

LLMs have demonstrated a remarkable ability to understand, interact with, and respond to human language Wang et al. (2018), Wang et al. (2019), OpenAI (2023), Brown et al. (2020). Nevertheless, understanding language or recalling facts alone does not guarantee understanding higher-order concepts or the construction of a world model Vafa et al. (2024) that describes and explains how things relate to one another. The discussion above highlights that for LLMs to demonstrate comprehension of high-level concepts, they must be able to discern similarities and differences between complex concepts. Consequently, measuring an LLM's proficiency in discerning similarity and difference within highly specific concepts and fields offers a powerful proxy for evaluating its understanding of the conceptual foundations and landscape of these fields.

Our measure of understanding draws inspiration from previous benchmarks used to assess language comprehension in earlier generations of language models. GLUE Wang et al. (2018), for example, includes tasks that involve determining whether two sentences convey similar meanings or whether one entails the other. Similarly, our tasks focus on whether models can distinguish subtle conceptual differences. While prior benchmarks often relied on synthetic or crowd-sourced data, e.g. Bowman

et al. (2015), our use of patent abstracts introduces a more challenging and high-stakes setting, requiring a fine-grained understanding of technical content.

## B    RELATED LITERATURE

**Relations to other benchmarks**: LLMs have achieved impressive scores on many NLP benchmarks, often giving the illusion of human-level reasoning or understanding. But researchers caution that excelling at a benchmark is not equivalent to possessing the general ability it is named after (McIntosh et al., 2024; Wu et al., 2024; Davis, 2023). High performance may reflect superficial pattern matching or memorization rather than true conceptual comprehension. In particular, current evaluation datasets frequently allow models to exploit statistical associations or training data overlaps to get the right answers without genuine understanding [5]. Even when test content is novel, models can exploit formatting quirks. Alzahrani et al. (Alzahrani et al., 2024) showed that minor changes to the evaluation format – like shuffling multiple-choice answer order – can significantly alter MMLU performance and even flip model leaderboard rankings by several positions. Such brittleness indicates models are learning task-specific tricks (e.g., defaulting to a particular choice position or prompt wording) rather than demonstrating robust knowledge application. Minor wording tweaks could cause inconsistent outputs, suggesting that models are latching onto spurious lexical cues instead of grasping underlying semantics (Arakelyan et al., 2024). Aggregate metrics in existing benchmarks often obfuscate key information about where models tend to succeed or fail, broadcasting an inflated view of LLM capabilities (Burnell et al., 2023). In sum, many current benchmarks are static and shortcut-prone, failing to truly assess whether an AI understands concepts or is simply leveraging surface regularities. Recognizing these deficiencies, researchers have proposed more diagnostic evaluations to probe the conceptual depth of LLMs, such as contrastive evaluation using paraphrase (Asthana et al., 2024) or contrast and counterfactual sets (Lewis and Mitchell, 2024) [6].

**Organizing internal and external knowledge**: A core challenge in LLM understanding is how to activate the vast latent knowledge embedded in the model's billions of parameters. Modern LLMs are trained on enormous corpora and ostensibly "know" a great deal, but this knowledge is stored in a highly compressed, distributed form not always accessible on demand (Borgeaud et al., 2022; Hoang et al.; 2023; Jaiswal et al., 2024; Pan et al., 2025). Indeed, there is a tension: an LLM may contain the facts needed for a task but still fail to recall them or apply them correctly when prompted. The model might skillfully "reason" with the information it does recall, yet still hallucinate a critical missing fact or overlook a subtle piece of world knowledge. This limitation is evident in tasks requiring niche or up-to-date information: the needed knowledge might exist somewhere in the model's parameters, but eliciting it reliably through prompts is non-trivial (Schulhoff et al., 2024; Gopnik and Wellman, 2012). To address these issues, researchers have proposed retrieval-augmented generation (RAG) techniques, which marry LLMs with external knowledge sources. Rather than relying solely on the model's compressed memory, a retrieval module fetches relevant documents (e.g., from Wikipedia or scientific literature) to provide the model with explicit evidence (Prince et al., 2024). Lewis et al. (Lewis et al., 2020) demonstrated the promise of this approach by generating more specific, diverse and factual language than a state-of-the-art parametric-only seq2seq baseline in NLP tasks. Borgeaud et al. (Borgeaud et al., 2022) introduced the RETRO (Retrieval-Enhanced Transformer) model. A RETRO model of only 7 billion parameters matched the performance of a 178B parameter model without retrieval. He et al. (He et al., 2022) showed that an LLM with a retrieval step produces more faithful, grounded understanding – their "rethinking with retrieval" method led to answers that stayed consistent with retrieved evidence, improving transparency and correctness. In general, augmenting LLMs with retrieval has been found to boost factual accuracy, reduce hallucinations, and expand the range of answerable queries. This is likely because LLMs gain new information during retrieval. LLMs compress information, and so lose rare information in the "tail" of the distribution, leading to phenomena like model collapse when trained on recursively generated data (Shumailov et al., 2024). By searching through uncompressed information with RAG, LLMs also access critical and potentially rare information for making detailed decisions about complex concept similarities and differences.

**Contrastive learning**: Our measures are also related to recent work on contrastive learning, which aims to improve model representations by explicitly training them to recognize semantic similarity and dissimilarity (He et al., 2020; Gao et al., 2021; Chen et al., 2020; Cohan et al., 2020). Contrastive methods work by pulling representations of semantically related inputs closer together while pushing

---

[5] https://www.sciencenews.org/article/ai-understanding-reasoning-skill-assess

[6] Some have expressed concern that even discriminative capacity falls prey to the "Generative AI Paradox", wherein models can produce fluent, even expert-level outputs that far exceed their actual comprehension of the content (West et al., 2024).

those of unrelated inputs apart, fostering a latent space that reflects nuanced conceptual structure. While our evaluation does not involve contrastive training *per se*, it is motivated by a similar principle: if a model can consistently discern between distinct concepts, it demonstrates a deeper form of understanding—one likely to translate into stronger performance on downstream tasks.

**Self-questioning and self-growth of LLMs**: LLMs have recently been prompted to engage in self-questioning and introspection to improve their reasoning and conceptual understanding (Huang et al., 2023; Qu et al., 2024; Song et al., 2024; Wang et al., 2023c;b). Instead of only generating direct answers, many contemporary models are encouraged to ask themselves clarifying questions and work through sub-problems, mimicking a Socratic or pedagogical approach. Early prompting techniques like chain-of-thought (CoT) (Wei et al., 2022) demonstrated that guiding LLMs to reason step-by-step markedly boosts performance on complex tasks. Building on this, researchers introduced methods for the model to generate its own intermediate questions. For example, self-ask prompting (Press et al., 2022) explicitly guides the model to pose follow-up questions to itself and then answer them in sequence, effectively decomposing complex queries into tractable pieces. Beyond decomposition, recursive self-reflection techniques have emerged to let LLMs iteratively improve their answers. Madaan et al. (Madaan et al., 2023) introduce Self-Refine, a framework where an LLM generates an initial answer and then critiques and revises its own output in multiple rounds. Shinn et al. (Shinn et al., 2023) propose Reflexion, in which an agent-model "verbally reflects" on feedback from previous trials and stores these self-generated insights in memory to inform subsequent attempts. Huang et al. (Huang et al., 2023) showed that even without external labels, an LLM can "self-train" by generating reasoning traces for unsolved problems and fine-tuning on them, yielding notable gains in mathematical and commonsense tasks. Miao et al. present SelfCheck (Miao et al., 2024), a zero-shot verification schema enabling LLMs to assess their own step-by-step reasoning without external resources, thereby enhancing accuracy in complex tasks through internal consistency checks. Similarly, techniques like least-to-most prompting (Zhou et al., 2022) encourage the model to teach itself gradually, which has improved performance on multi-step reasoning challenges. Other work has explored letting models self-evaluate or re-rank their answers (e.g., self-consistency voting (Wang et al., 2023a)) and prompts them to explicitly verify or justify their solutions, analogous to a student checking their work. These introspective or pedagogically-inspired strategies are motivated by human learning: students often learn better by asking themselves questions and reflecting on mistakes. Empirically, such approaches can bolster an LLM's conceptual grasp and reduce reasoning errors. Nevertheless, there are also limitations and mixed findings. It is hard to evaluate the quality of self-generated questions because LLMs may not be able to answer their self-generated questions (Balepur et al., 2025). As we demonstrate, large models are much more likely to productively answer small model questions than the converse. Not all tasks benefit uniformly from self-reflection. Li et al. (Li et al., 2024) find that prompting GPT-4 to reflect on its answers helps on a truth-focused QA task but hinders performance on a multi-hop knowledge task. Thus, while self-questioning and introspective prompts represent a powerful paradigm for enhancing LLM reasoning, careful tuning is needed to ensure the model's "inner dialogue" remains productive and does not overthink or hallucinate.

# C    JUDGING CLOSE PATENTS VS. REWRITTEN PATENTS

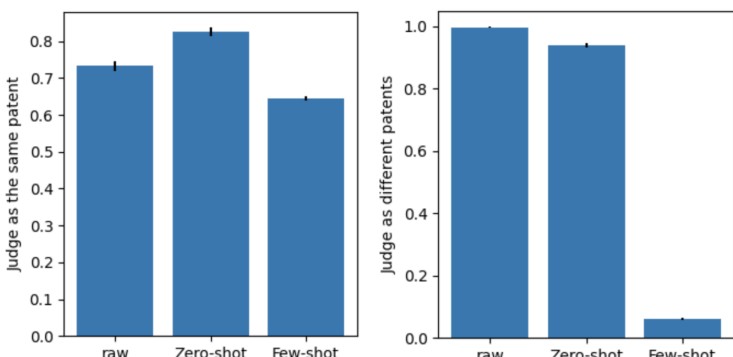

(a) Few-shot CoT fails to help distin-
guish between patents with meaningful
technical differences.

(b) Few-shot CoT helps identify simi-
larities across patent abstracts and their
rewritten versions.

Figure 6: Few-shot CoT reasoning can distinguish the rephrased abstracts (rewritten by the GPT-o3 model), but cannot distinguish their counterpart (another unique patent).

Following on prior sections that focus on distinguishing deceptively similar patents, in this section, we explore whether and how LLMs are able to merge identical but paraphrased patent ideas. Our approach to measuring LLMs' ability to generalize and identify the same ideas across distinct articulations is to generate rephrased versions of each patent abstract using the OpenAI ChatGPT o3 model with the prompt "rephrase the patent, but keep its meaning." We then task the LLaMA-3.1-8B-instruct model with identifying whether a patent abstract and its paraphrase describe the same invention. This was designed to parallel the original distinguishing experiment, in which we randomly sampled 1,000 patents and identified pairs that were most semantically similar. We then asked the LLaMA-3.1-8B-instruct model to determine whether the two abstracts described the same invention, using a human-annotated ground truth indicating that they were different. In raw (zero-shot) judgments, the model performed poorly on both merging and splitting tasks. We then introduced zero-shot CoT prompting ("think step-by-step") and few-shot CoT, where we supplied examples of both merging (paraphrased) and splitting (semantically similar but distinct) patent pairs. Few-shot CoT substantially improved the model's accuracy in recognizing paraphrased abstracts as equivalent—but had minimal effect on its ability to distinguish between closely related yet different inventions.

This pattern suggests that CoT reasoning helps the model overcome basic semantic similarity checks and identify paraphrases, likely by encouraging step-by-step alignment at the surface level. Our hypothesis is that the model passes the rephrasing test primarily through pattern matching and semantic overlap, rather than through true conceptual integration. This explains why CoT improves performance on the rephrasing task: the test is relatively easy in the sense that our rewritten version keeps semantic similarity high, such that shallow strategies suffice. In contrast, differentiating between semantically similar but substantively different patents remains hard, even with CoT, because it requires more than aligned phrasing—it demands conceptual depth.

In principle, a more advanced rephrasing model—one that intentionally avoids superficial alignment and instead rewrites concepts in deeper, structure-preserving ways—might push the LLM closer to the more difficult task of distinguishing near matches. Designing such a rephrasing system is far from trivial. Thus, while our current rewrite approach does not fully obscure surface similarity, the model's success only when the task is easy reinforces the motivation behind our benchmark: LLMs rely heavily on surface-level semantic similarity, rather than demonstrating a deeper understanding of the underlying concepts. In this way, our dataset represents a frontier capability for all contemporary LLMs, even though larger models outperform smaller ones.

# D  THE PATENT DATASET

The patent dataset comprises 1,319,184 patent pairs in total, drawn from five Cooperative Patent Classification (CPC) classes: H05 – Electric techniques not otherwise provided for; H04 – Electric communication technique; G06 – Computing; Calculating; Counting; H03 – Basic electronic circuitry; and G11 – Information storage. Within these, the patents are further categorized into 41 CPC subclasses: ['H05B', 'H04L', 'H05H', 'H05K', 'G06V', 'G06N', 'G06T', 'H04N', 'G06F', 'G06Q', 'H03M', 'G06K', 'H04B', 'H04R', 'G11B', 'G11C', 'H04M', 'H03B', 'H03F', 'H03G', 'H03H', 'H03K', 'H03L', 'H04H', 'H04J', 'H04K', 'H04W', 'H04Q', 'H04S', 'H05F', 'H05G', 'H03C', 'H03J', 'G06E', 'H03D', 'G06G', 'G06M', 'G06C', 'H05C', 'G06J', 'G06D']. Out of all matched patent pairs in our dataset, 82.55% share the same CPC class, and 68.10% share the same, more specific CPC subclass. In contrast, only 10.50% of the pairs were granted in the same year.

This is an example patent pair where all LLaMA 3.1 instruct models — 8B, 70B, and 405B - fail to distinguish. We also show a scientific paper abstract to help readers perceive the distinct writing styles between patents and papers.

**Patent 1**: *A wireless device (18) is configured for use in a wireless communication system. The wireless device (18) is configured to receive, from a radio network node (12), a multicast or broadcast transmission (14) and downlink control information indicating a number of repetitions with which the multicast or broadcast transmission (14) is transmitted. The wireless device (18) is also configured to operate according to a negative acknowledgement only feedback scheme in which the wireless device (18) is configured to transmit a negative acknowledgement to the radio network node (12) if decoding of the multicast or broadcast transmission (14) fails using the indicated number of repetitions and to refrain from transmitting a positive acknowledgement to the radio network node (12) if decoding of the multicast or broadcast transmission (14) succeeds.*

**Patent 2**: *Methods, systems, and devices for wireless communications are described that support feedback for multicast communications. A user equipment (UE) in a group of UEs may receive multicast information from a base station in a downlink (DL) transmission scheduled by a DL grant. The DL grant may also indicate a set of uplink resources for the group of UEs to use to transmit feedback. The UE may attempt to decode a message containing the multicast information in the DL transmission. If the UE determines the message was successfully decoded, the UE may send no feedback to the base station. If the UE determines the message was not successfully decoded, the UE may transmit a feedback message to the base station to indicate the message was unsuccessfully decoded. The base station may monitor for the feedback message and determine to retransmit the multicast information if the feedback message is received.*

**An Arxiv paper including the answer from the sampled patent pair**: *The Third Generation Partnership Project (3GPP) has recently published its Release 16 that includes the first Vehicleto-Everything (V2X) standard based on the 5G New Radio (NR) air interface. 5G NR V2X introduces advanced functionalities on top of the 5G NR air interface to support connected and automated driving use cases with stringent requirements. This paper presents an in-depth tutorial of the 3GPP Release 16 5G NR V2X standard for V2X communications, with a particular focus on the sidelink, since it is the most significant part of 5G NR V2X. The main part of the paper is an in-depth treatment of the key aspects of 5G NR V2X: the physical layer, the resource allocation, the quality of service management, the enhancements introduced to the Uu interface and the mobility management for V2N (Vehicle to Network) communications, as well as the co-existence mechanisms between 5G NR V2X and LTE V2X. We also review the use cases, the system architecture, and describe the evaluation methodology and simulation assumptions for 5G NR V2X. Finally, we provide an outlook on possible 5G NR V2X enhancements, including those identified within Release 17.*

# E  PROMPTS

---

**Prompt for Question Generation**

**System Prompt:** You are an expert in patent comprehension. Your task is to generate structured questions that assess a reader's background knowledge necessary to understand a given patent abstract. The questions should focus on foundational concepts, principles, and applications relevant to the patent's domain without explicitly referencing the patent itself. Ensure the questions are framed generally and test for domain knowledge rather than details specific to the patent.

The questions should follow Bloom's Taxonomy and be categorized into three levels:

1. Remembering: Questions that assess the reader's ability to recall key technical concepts, definitions, and fundamental principles relevant to the patent's domain. These should focus on identifying and defining key terms, recognizing fundamental components, and understanding core principles of related technologies.

2. Understanding: Questions that assess the reader's ability to explain how different elements of similar technologies function and interact. These should encourage summarization, interpretation, and explanation of related technical concepts and their roles in broader systems.

3. Applying: Questions that evaluate the reader's ability to apply their knowledge by solving problems, making predictions, or considering real-world applications of the broader technological principles underlying the invention. These questions should be scenario-based and encourage practical thinking.

Do not directly reference the patent abstract in any question. Instead, ensure all questions test knowledge that would be useful for understanding the patent without explicitly addressing its content. Maintain a structured and logical flow from basic recall to deeper conceptual understanding and practical application.

**Basic Level Question Prompt:**

You are given the following patent:

Patent Title: {patent_title}

Patent Abstract: {patent_summary}

Generate a set of {qnum} questions that test a reader's ability to recall fundamental concepts, key terms, and basic components necessary to understand the given patent abstract. The questions should focus on defining terms, identifying parts, and recognizing the primary function of the described technology.

Output Format (JSON):

Your response should be formatted as valid JSON:

{{
"1": "question1",
"2": "question2",
"3": "question3"
}}

**Conceptual Level Question Prompt:**

You are given the following patent:

Patent Title: {patent_title}

Patent Abstract: {patent_summary}

Generate a set of {qnum} questions that assess a reader's comprehension of the given patent abstract by requiring them to explain how different elements of the invention work together. The questions should encourage the reader to summarize the design, describe interactions between components, and interpret the intended improvements of the invention.

Output Format (JSON):

Your response should be formatted as valid JSON:

{{
"1": "question1",
"2": "question2",
"3": "question3"
}}

---

**Prompt for Answering Questions with Scientific Knowledge**

You are a scientific reasoning assistant.
Use the following context to answer the question. Focus only on the information provided.
Context: // retrieved scientific knowledge
{context1}
{context2}
{context3}
Question:
{question}
1. Identify and summarize key points from the context relevant to the question.
2. Synthesize those points with your knowledge and try your best to answer the question.
Return your answer in the following JSON format:
{{
"answer": "your answer" //your answer to the question
}}

**Prompt for the Splitting Task Judgment**

You are a patent judge. Below are two patent abstracts with extra information.
This extra information may or may not be useful in assessing the patents. You may choose to use it at your discretion. // This is for placebo answers
Your task: judge whether they describe the same patent.
Abstract 1:
{abstract1}
{qa_block1}
Abstract 2:
{abstract2}
{qa_block2}
Strictly return your concise answer in the following JSON format:
{{
"label": 1 // 1 if they describe the same patent, 0 if not
"confidence": 10 // your confidence score from 0 to 10, with 10 representing the highest confidence
}}
// Note:
// This applies to the reverse questioning scenario (the major experiments in the paper):
// "Do you think they are the same patent?" — when, in fact, the patents are judged to be *different* by human experts.
// Invert this setup: Replace "same" with "different", and return 1 if the LLM concludes that the patents are *different*. This yields an alternative version of the experiment (See Section 3 Figure 2 in the paper).

**Prompt for Self-talk Answering**

Try to answer the question.
Question: {question}
Return your concise answer in the following JSON format:
{{
"answer": "your answer" // your answer to the question
}}

# F RESULTS FOR MAJORITY VOTING SCHEMA

## F.1 PERFORMANCES OF OTHER LLMS WITH VARYING SIZES ON SPLITTING PATENTS

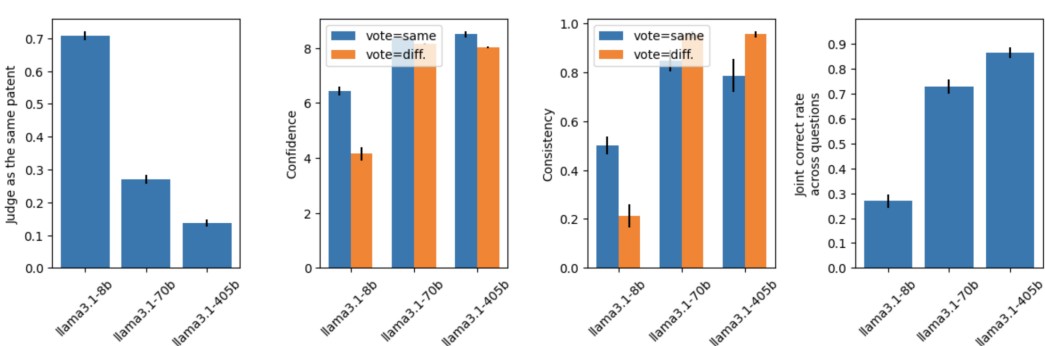

Figure 7: Ask LLMs "Do you think they are the same patent?", given that they are actually different. Judge using majority voting, and 1 = same. Models are instruction-tuning versions.

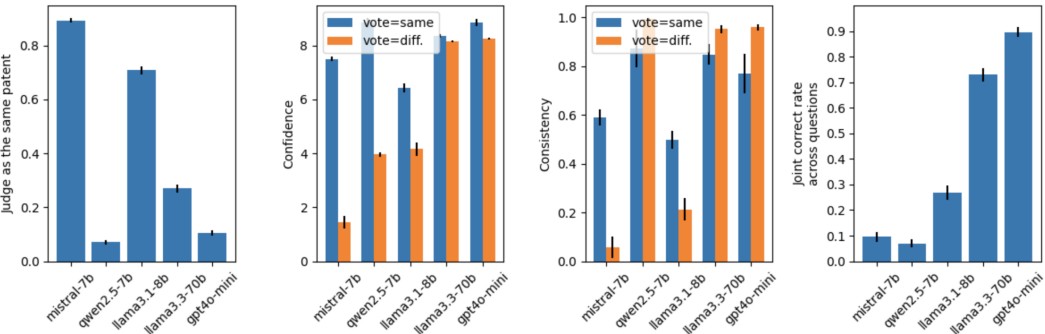

Figure 8: Ask LLMs "Do you think they are the same patent?", given that they are actually different. Judge using majority voting, and 1 = same. Models except for GPT-4o-mini are instruction-tuning versions.

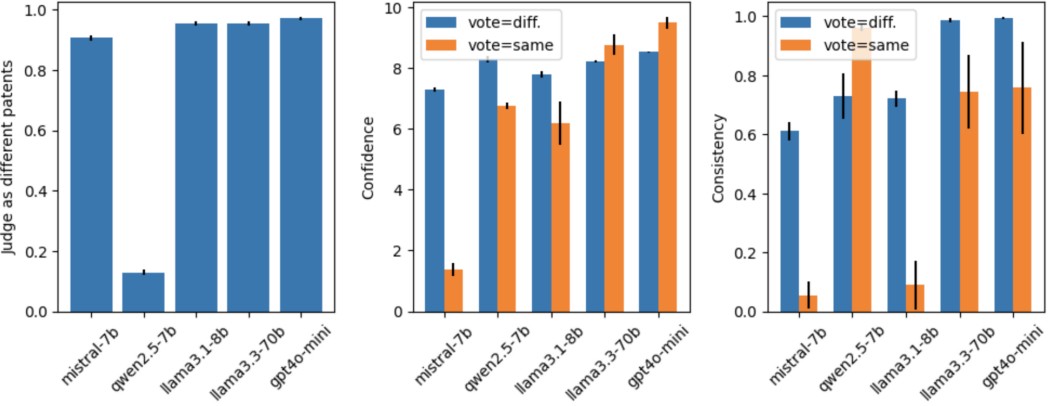

Figure 9: Ask LLMs "Do you think they are different patents?", given that they are actually different. Judge using majority voting, and 1 = yes. Models except for GPT-4o-mini are instruction-tuning versions.

## F.2  LLMS DO NOT KNOW WHAT THEY HAVE KNOWN

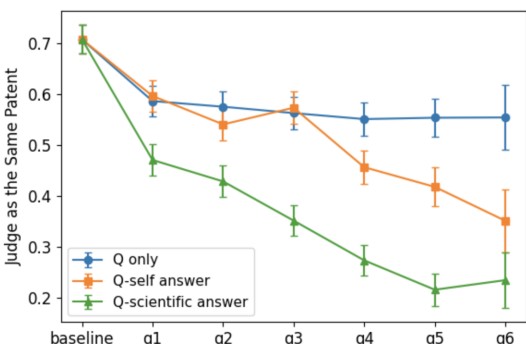

Figure 10: Asking Llama 3.1 8B instruct "Do you think they are the same patent?" on actually different patents (1 = wrong answer "yes"). LLM answers using **majority voting**. It does not know what it has known. These results are statistically significant: Self-talk still helps improve the baseline judge but not as good as scientific self-questioning: $q_1$ to $q_5$ p=0.000; $q_6$ p=0.006.

### F.3 SMALLER MODELS' QUESTIONS HELP MID-SIZE MODELS MORE THAN THOSE FROM LARGER MODELS

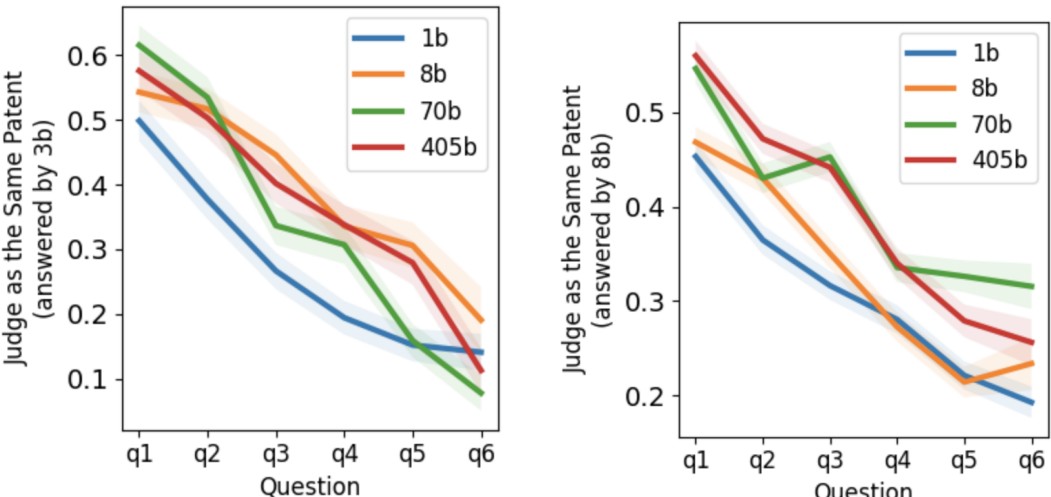

Figure 11: **Llama 3B/8B's judge with questions proposed by different models with different sizes.** Ask "Do you think they are the same patent?", with the ground truth being *different*. We observe that bigger models' questions are less useful for the answer models to judge.

## G DISCUSSION OF LIMITATIONS AND FUTURE WORK

Several limitations and directions for future work remain.

Although our current setup uses scientific retrieval from arXiv papers, a more promising direction is to evolve the retrieval process itself. Rather than relying on fixed sources or searching questions from scientific databases, LLMs could learn to refine or even generate better retrieval queries via reinforcement learning, active learning, or model-in-the-loop optimization based on science or what they have learned from prior reasoning steps. An agentic framing - where models, like a baby sensing the world, autonomously refine their questions and knowledge, decide whether to retrieve information, evaluate whether to accept or reject what is retrieved, and determine how much knowledge to seek for one question/how many questions to ask - could enable deeper forms of exploration beyond single-pass self-questioning.

The exact nature of "resolution" in distinguishing patents remains underdefined. That is, even when models succeed at effectively splitting, it is not always clear what semantic threshold they are using. Are they resolving based on functionality, application, architecture, or terminology? Future work could explicitly model and evaluate these questions to understand what kinds of conceptual distinctions LLMs are truly capable of making by probing their internal, layer-wise conceptual representations and dynamics. It will also be interesting to look into differences in judgment performance across patent classes and categories, as LLM internal knowledge likely varies across fields.

