# OpenReview forum: "Automatically Advancing LLM Expertise in Technology Judgment"
_ICLR.cc/2026/Conference — ICLR 2026 Conference Withdrawn Submission_

### Official Review · Reviewer_PXH4 · 2025-10-25

**Soundness:** 2
**Presentation:** 2
**Contribution:** 2
**Rating:** 4
**Confidence:** 3

**Summary:**

This paper proposes a benchmark that consists of 1.3M+ patents, where the task is: given a pair of patent abstracts, the LLM has to decide if they're the same or different. Patents are chosen because they're inherently different from each other (one cannot get a patent approved if it already exists), and are somewhat ambiguously worded, making the task not easy. Results are reported across the LLaMa 3.1 family of models, wherein the 8B parameter version struggles and the 405B manages to correctly judge pairs at close to a 90\% accuracy rate. The paper investigates the errors made by LLMs on the benchmark to determine if they are due to 1) the models not reasoning enough about the given information or 2) the models not having enough information.

**Strengths:**

- The dataset is thorough (1.3M+ samples) and creative, featuring an out-of-the box domain
- The paper is easy to understand

**Weaknesses:**

I think one limitation of the paper is that it mainly evaluates LLaMa 3.1 models, which are not among the top models, even among open-source ones. Still, the best LLaMa model reaches nearly 90\% accuracy, and GPT-4o mini performs similarly. Given this, it seems likely that stronger models such as Qwen-32B or GPT-5 would exceed 90\%, which raises questions about how challenging the benchmark remains and, in turn, about its broader interest to the community.

In general, is there a reason why abstracts rather than full texts are used? Because abstracts do not contain all the patent information, could it be the case that for some pairs, it is impossible to decide if they differ based on just abstract text alone?

In addition, the analysis might have a confounding factor. When the model is given scientific papers related to the patent, it also receives more tokens and does more processing. How would the results change if, instead of related papers, the model were given the full patent text (as Knowl_{ij} in Figure 3)?  It would be helpful to see examples where the scientific context improved performance and what information made the difference.

Finally, if I understand correctly, in some cases the model is given the same patent. How is this handled? Is the text modified when patent_1 = patent_2 to make the question non-trivial?


Minor:
- line 249: Appendix should be capitalized
- the plots seem a bit cluttered and information dense. In Figure 5 for example, perhaps some of the plots could be put in the appendix, which would leave room for bigger plots, axes and labels, and more information in the caption. Also could be beneficial: despining the plots, making the legend backgrounds transparent, and having one legend instead of two in the case of Figure 2 (because the legends are the same). Also: the individual captions under the plots (for example, Figure 1) could be removed in favor of a longer, more in depth general caption, like there is at the bottom of Figure 1.

**Questions:**

1. How would the results change if GPT-5, Qwen32B or Claude Sonnet 4.5 were used? Because these models are more expensive to run, results on a subset of the benchmark could be significant here.

2. How would results change if the full text of the patent was used and not just the abstracts? How would the self-answer/scientific analysis change if the patent text was used in place of $Knowl_{i,j}$?

3. When scientific context does help reduce errors, would it be possible to get examples of the missing knowledge that helped the LLM come to a decision?

---

### Official Review · Reviewer_GGwd · 2025-10-27

**Soundness:** 2
**Presentation:** 2
**Contribution:** 3
**Rating:** 2
**Confidence:** 4

**Summary:**

This paper proposes to assess whether LMs truly use their knowledge when confronted with new and challenging tasks. To address this problem, they collected a challenging new benchmark of 1.3 million CompSci patent pairs. With a novel framework that decomposes model erros into two sources (missing and unused knowledge), this proposed approach asks models to generate clarifying questions. This self-question framework highlights a critical limitation of current LLMs: models often know more than they can use.

**Strengths:**

- The patent dataset is large, and would be useful for compsci researchers, including but not limited to LLM developers.
- The research question asking whether LLMs truly use their knowledge when confronted with new and challenging tasks is valuable.

**Weaknesses:**

- While the research question (whether LLMs truly use their knowledge when confronted with new and challenging tasks) and the solution (examine with a dataset involving patent judgment) are both valuable, they do not seem to me to have the same scope for a paper. I highly recommend that the authors reconsider the scope of the research question in the introduction: instead of asking "whether LLMs can enhance their own understanding", ask "whether LLMs can perform the technology judgment task that involves distinguishing patents that are near duplicates". This would require rewriting the abstract and the introduction, etc., but the revised story will have a more specific scope and will therefore fit the empirical contribution better. Following this framing, the "self-questioning" can be established as a proposed solution. This should be compared to several baselines (e.g., Transformer-based classification, prompt-based classification). I think centering on the dataset is an easier way to improve this paper. (The alternative is to use additional datasets that answer the question "whether LLMs can enhance their own understanding" more thoroughly -- but given the vagueness of this question, I prefer the current recommendation to the authors.)
- Several other mismatches between the research question and the research evidence should be fixed by more careful wording. For example, "how LLMs' internal/external knowledge are organized and transferred across model scales" is not studied thoroughly enough in the experiments. The current experiment, on the other hand, studies "how LLMs across different scales have knowledge about technology judgment". Note how this subtle difference in choice of words reframes the research question.

**Questions:**

Aside: A previous version of this paper was in my batch at another conference a few months ago. I see this version has made some edits, but still, this version is not good enough.

---

### Official Review · Reviewer_1WND · 2025-10-28

**Soundness:** 2
**Presentation:** 2
**Contribution:** 2
**Rating:** 4
**Confidence:** 4

**Summary:**

This work introduces a new dataset of over 1.3 million close-but-distinct computer science patent pairs for patent classification task. Using this dataset, it investigates two error sources (missing or unused knowledge) by comparing performance across three settings: raw performance, self-answered questions, and externally supplied answers. The results show that LLMs often possess the relevant knowledge internally but fail to deploy it. Further analysis shows that smaller models generate simpler, broadly transferable questions, while larger models propose more complex but less generalizable ones.

**Strengths:**

- The proposed patent dataset appears highly valuable for LLM evaluation. As noted by the authors, patents are hard to read and understand, hard to distinguish from one another, and often strategically obfuscated. This makes them an important and valuable source within broad technological domains.

- The idea that “a central aspect of understanding is differentiation” is interesting.

- The dataset’s scale is impressive, encompassing more than 1.3 million patents.

**Weaknesses:**

- My primary concern is that the current task formulation, based on binary classification, appears too simple. Several models, including Llama 3.1-405B, Qwen2.5-7B, Mistral-7B, and Llama 3.1-8B, already achieve close to 0.90 accuracy under simple zero-shot prompting (as reported in Appendix F). Some of these are not the latest models, suggesting the task may be insufficiently challenging and potentially very sensitive to randomness. It would strengthen the benchmark to explore more complex evaluation setups, such as requiring models to articulate the differences between two patents or to simulate expert-style examination reports. Including tasks across varying difficulty levels could make the benchmark both harder and more comprehensive, thereby increasing its practical utility.

- The plots are confusing, particularly due to inconsistent and unclear y-axis labeling. Replacing labels such as “judge as the same patent” or “judge as the different patent” with standard performance metrics (e.g., accuracy) would greatly improve readability. Several other figures could also benefit from clearer organization and visual consistency.

- In Figure 1(b), please include comparison data from non-patent documents. Without such context, it is difficult for readers to assess whether patents are similar.

- The main body only reports results for Llama 3.1, while results for other models are relegated to the appendix. Results from one model are not convincing and I highly recommend moving other results to the main paper.

- The conclusion that “LLMs do not know what they know” has been widely reported in prior work. While the presented results provide additional evidence to that, I am looking for something new and interesting.

**Questions:**

- It may be worth exploring a setting without predefined question types, allowing models to freely generate questions.

- In Appendix D, some examples include numbered elements such as “(18), (12), (14).” Could the authors clarify whether referenced evidence associated with these numbered components are included in the input? If so, does this substantially increase the context length, and how do the authors manage context window constraints?

---

### Official Review · Reviewer_CuVx · 2025-10-31

**Soundness:** 3
**Presentation:** 2
**Contribution:** 2
**Rating:** 4
**Confidence:** 2

**Summary:**

The paper introduces (i) a benchmark and a training dataset for the evaluation and fine-tuning of large language models on the scientific-to-policy brief generation and (ii) a corpus of science-policy pairs through a multi-step filtering and in-context polishing pipeline, to fine-tune three open-source models.

**Strengths:**

- The task addressed in the paper is (as far as I see) original and scientifically relevant
- The proposed taxonomy is well structured
- The pipeline to define and build the dataset is well detailed
- The experimental section comprises a large number of LLMs

**Weaknesses:**

The introductory motivation could be improved in the following phase  “LLMs struggle with policy brief generation” (page 1), which is used as a core motivation but is not contextualized with quantitative evidence or prior benchmarks. It would strengthen the introduction to include a preliminary reference or citation illustrating the current performance gap that motivates the work.

The selected venues (Nature Energy, Nature Climate Change, Nature Cities, Nature Sustainability, and JHSB) seem to focus on environmental and social sciences. While this is a valid and important focus, the approach’s generalizability to other domains, such as engineering, AI, or technological innovation, remains unclear. A discussion of cross-domain applicability would be helpful.

The reliance on an LLM-as-a-judge evaluation approach is both innovative and potentially fragile.
Since the fine-tuned models and the evaluation models may share similar biases, the LLM-as-a-judge method risks circularity. The authors mention mitigation strategies in appendix , but this remains a critical methodological assumption that should be discussed more in depth.

The experimental section is large and full of analysis. Unfortunately, in the opinion of this reviewer, a more detailed set of comments about those numbers would have helped to gain more insights.

Some figures (e.g., Fig. 1) could be more clearly annotated, and minor typographical issues persist. A tighter organization of tables and figures would improve readability. The acronym JHSB (Journal of Health and Social Behavior) appears in Figure 1b but is only defined later in the paper (page 23), which disrupts readability.

**Questions:**

See previous box

---

### Note · Authors · 2025-11-15

I have read and agree with the venue's withdrawal policy on behalf of myself and my co-authors.